# Minimal Cardiac Perforation by Lead Pacemaker Complicated with Pericardial Effusion and Impending Tamponade: Optimal Management with No Pericardiocentesis Driven by Echocardiography

**DOI:** 10.3390/diagnostics10040191

**Published:** 2020-03-30

**Authors:** Carlo Caiati, Paolo Pollice, Luigi Truncellito, Mario Erminio Lepera, Stefano Favale

**Affiliations:** 1Institute of Cardiovascular Disease, Department of Emergency and Organ Transplantations, University of Bari “Aldo Moro”, 70121 Bari, Italy; paolo.pollice@yahoo.it (P.P.); marioerminio.lepera@uniba.it (M.E.L.); stefano.favale@uniba.it (S.F.); 2Unit of Cardiology, Civil Hospital “Giovanni Paolo II”, 75025 Policoro (MT), Italy; luigi.truncellito@asmbasilicata.it

**Keywords:** lead complications, cardiac tamponade, echocardiography

## Abstract

We report the case of a 51-year-old patient who underwent the implantation of a bi-ventricular implantable cardioverter defibrillator (ICD) complicated by a sub-acute right ventricular minimal perforation with pericardial effusion and echocardiographic signs of tamponade. A new echocardiographic plane orientation allowed us to diagnose this condition in emergency and to make the right decision without delay, which consisting in unscrewing the active fixation screw under fluoroscopy guidance, while the pericardiocentesis was postponed. Thanks to the intervention focused on eliminating the cause of the postcardiac injury syndrome, the patient recovered rapidly and ultimately avoided the pericardiocentesis procedure.

## 1. Introduction

Cardiac perforation (both atrial or ventricular cavity) by pacemaker and cardioverter defibrillator leads is, unfortunately, an escalating phenomenon [1], with an incidence of 0.6% (95 CI = 0.4% to 0.8%), as reported in a large population-based cohort study (Danish registry) [2], but this figure is higher if you consider the asymptomatic [3] perforation rates that were reported: 15% (9/61) for atrial and 6% (6/100) for ventricular leads.

This implies lead dislodgements and worrisome consequences like cardiac tamponade, which is handled with urgent pericardiocentesis [4]. It is extremely important to have a rapid, non-invasive—and possibly non-ionizing—radiation method, which is thus repeatable, for a prompt and precise diagnosis in the event of suspected lead perforation, which, ideally, should show us the precise extent of the problem and at the same time, document any other related complications, such as cardiac tamponade. Cardiac pericardiocentesis is a symptomatic treatment that is temporarily life-saving, but which implies peri-procedural risk (death, ventricular puncture, etc.), especially if performed in an emergency [5]. However, a precise detection of the cause of tamponade—if totally removed—may allow the rapid regression of the tamponade, thus avoiding pericardiocentesis.

We report a case of minimal right ventricular apex perforation of the tip (screw) of a ventricular active-fixation lead, with consequent pericardial irritation and pericardial effusion with impending tamponade (echocardiographic signs of tamponade without causing circulatory collapse, as indicated by blood pressure and perfusion status) [6]. Echocardiography with a modified transections plane was able to perfectly visualise the lead dislodgement. These diagnostic details provided a strategy for removing the cause and postponing the pericardiocentesis that was ultimately no longer necessary, as a natural, rapid recovery took place after the elimination of the irritative stimulus. 

## 2. Case

A 51 year-old man with ischemic dilated cardiomyopathy (DCM), already revascularized with a PTCA procedure on the left anterior descending (LAD) coronary artery, underwent the implantation of a three-chamber biventricular implantable-cardioverter-defibrillator (ICD) (ST Jude Medical Epic II + HF model V-357) due to the persistence of a significative symptomatic (NYHA class III) left ventricular systolic dysfunction (left ventricular ejection fraction 25% evaluated by the Simpson biplane method) associated with left bundle branch block (LBBB). The procedure was performed under local anaesthesia. The device was implanted in the sub-clavicular area and connected to the three leads. The right atrial bipolar passive-fixation lead (Medtronic 4592-53 cm, LER 150033V) was positioned via puncturing the left subclavian vein and positioned in the right atrial appendage. The right ventricular (RV) bipolar screw-in (active fixation) silicone lead (ST Jude 1580-65 cm, Riata) was inserted via the cephalic vein and positioned in the right ventricular apex. The left ventricular passive fixation lead (ST Jude, 1056T-86 cm QuickSite) was inserted via the jugular vein and positioned in the coronary sinus. Atrial and ventricular sensing and pacing thresholds were satisfactory. 

The device programming was (initial setting): 

Capture threshold: left ventricular voltage 0.4 × 0.5 ms; atrial lead voltage 0.2 × 0.5, right ventricular voltage 0.3 × 0.5 ms. 

Stimulation threshold: left ventricular lead voltage 5 × 0.5 ms; atrial lead 5 × 0.5 ms; right ventricular lead voltage 5 × 0.5 ms.

Impedance: Left ventricular lead 879 Ω; atrial lead 532 Ω; right ventricular lead 589 Ω.

Implantation was apparently uncomplicated. After 48 h from the ICD implantation, the patient was discharged asymptomatic; during the chest radiography and echocardiography performed after implantation, no complication was evident. Some hours after discharge, the patient began to complain of atypical chest pain. An ECG performed in the emergency department showed left ventricular hypertrophy and new negative T waves in D1, aVl, V5 and V6. A chest-x ray revealed a lung hilum enlarged with a possible left pleural effusion. The echocardiogram showed a moderate pericardial effusion (1.3 cm max diastolic pericardial layer separation). Laboratory markers were: Hb 10.1 g/dl, rbc 3.330.000, wbc 7580., and Hct 29.3%. The patient was therefore re-admitted to our unit, and in the ensuing days showed a clinical improvement, with a pericardial effusion reduction (0.8–1 cm diastolic pericardial layers separation), and he was then discharged. A few days later, he was admitted once again for an ICD inappropriate shock, due to an atrial flutter with high ventricular response; the echocardiographic examination showed little increase in pericardial effusion (1.6 cm max diastolic pericardial layers separation). A cardiac injury syndrome by lead irritation was hypothesized, and the patient was discharged with the suggestion of a subsequent elective control. 

A few days after discharge, the patient returned to the emergency department complaining of atypical chest pain, shortness of breath, and asthenia. Clinical evaluation revealed regular tachycardia at 145 b/m, blood pressure at 110–70 mmHg, no recognizable jugular waves, and jugular engorgement (the vertical distance of the top jugular column from the angle of Louis = 5–6 cm at 45° chest elevation) with positive markedly positive jugular reflux, mild pulsus paradox (5 mmHg), and no palpable apex beat. An EKG showed atrial flutter, with an atrio-ventricular conduction ratio of 2:1. The pacing parameters were within a normal range value, and substantially unchanged with respect to the baseline value. Echocardiography showed a further worsening of the pericardial effusion (2.5–3 cm diastolic pericardial layers separation), with echocardiographic signs of tamponade (Figure 1). 

The patient was thus scheduled for urgent pericardiocentesis; before the procedure, a new echocardiographic examination, specifically aimed at best visualizing the tip of the catheters, was performed (Figure 2). A four-chamber view with a sub-apical approach, in order to transect the true apex, allowed a full visualization of the ventricular lead to the apex (Figure 2) and by a slight downward inclination of the tomographic plane (as schematically shown in the Figure 2), we focused on the tip of the ventricular lead that had unequivocally penetrated into the pericardial space by a few mm. On the basis of knowledge of the Riata lead structure (Figure 2), it was evident that the visualized displaced ‘tip’ was mostly the fixating screw (helix) of the Riata lead (Figure 2). 

As the cause of the pericardial effusion with impending tamponade was quite clear, we decided to cancel the pericardiocentesis procedure and first of all remove the cause of the pericardial irritating stimulus and in the meantime, temporarily treat the congestion symptoms with diuretic therapy; this was simply attained by carefully unscrewing the fixating screw (i.e., the extendable/retractable helix for fixation in the ventricle (Figure 2)), along with minimal retraction of the lead, under fluoroscopic monitoring. That procedure was successful and uncomplicated. The same echocardiographic view used for the diagnosis showed the disappearance of the helix from the pericardial space and the visualization of the true tip of the catheter at the right ventricular apex, with no signs of perforation; the tip appeared as a hyperechogenic short area, maximally distally located (the tip electrode), separated by a few mm from another, more proximal, similar hyperechogenic area, that should be related to the ring electrode; also, the distal coil was distinguishable via the long hyper-echogenic area (Figure 2). A successful electrical cardioversion was also performed. In the following days, we observed a clinical and echocardiographic improvement, with complete regression of pericardial effusion on the 20th day (Figure 1). The clinical conditions remained stable, with no relapse of pericardial effusion at the 1-year follow-up.

## 3. Discussion

The major finding in our case is that a precise diagnosis with echocardiography allowed pericardiocentesis to be avoided. As a general and golden rule in medicine, the elimination of the cause(s) brings about the healing process [7]. Thus, it is necessary to make a considerable effort to find out all the causes of a malady [7]. In particular, echocardiography was able to visualize the helix (fixation screw) minimally protruding into the fluid-filled pericardial space; that prompted us to retract it by simply unscrewing it (the helix is retractable), with minimal disturbance for the patient, minimal fluoroscopy time and most of all, minimal postponement of the pericardiocentesis. This simple and innocuous unscrewing procedure totally eliminated the myocardium-irritating action of the lead and prompted rapid healing with total, swift regression of the effusion, with no need for pericardiocentesis. 

Echocardiography played a pivotal role in our case. In our view, echocardiography has great potential for handling patients with possible lead complications. This technique has the advantage of being rapidly available at the bedside, a quality which is maximally appreciated in emergencies like our case; it is the principal tool in diagnosing effusion and cardiac tamponade, which are major complications of lead perforation [8]; in addition, it is a non-ionizing radiation method that can afford repeatability without harming the patient, which is pivotal in a clinical situation like ours, where follow-up is crucial. In our case, the prompt recognition of minimal perforation was possible thanks to off-axis tomographic planes (Figure 2). Therefore, a thorough exploration of the catheter course by ultrasound, looking for tip visualization, is important. To orient oneself in lead imaging, very useful internal points of reference are the coil and the electrode, which appear as hyperechoic rings or areas along the catheter course (Figure 2). In our case, the helix, the two rings and eventually, the distal defibrillator electrode were visualized. We found it useful to know a priori the structure of the lead that we were going to explore by echo (Figure 2). The potential of echocardiography in lead imaging is increasing. As recently demonstrated, echocardiography (and in particular intracardiac ultrasound) has the potential to detect the fibrosis encapsulating the lead that brings about complications after infected lead extraction (the formation of new post-extraction masses) [9]. The value of this echocardiographic approach in this setting is also emphasized by the fact that impedance and other electrophysiological parameters didn’t change. A normal impedance doesn’t exclude lead dislocation with minimal cardiac wall perforation [10]. A small perforation, in fact, might result in the cathode being proximal to the epicardium and the anode being proximal to or within the endocardium, resulting in normal pacing parameters [10], as eventually happened in our patient, in whom only the helix clearly reached the pericardial space (Figure 2).

In other cases in the literature, late cardiac perforations were diagnosed with echocardiography: in one case (one month after implantation), a passive tined lead caused an important perforation, with the tip moving freely in the mediastinum, crossing the myocardium [11]; in another case, a very late (10 months) lead perforation from a Riata catheter with no pericardial effusion was correctly identified with echocardiography, and the Computed Tomography (CT) was used only to confirm the diagnosis [12]. 

Considering that most patients could be initially asymptomatic [3], TTE could be considered an appealing screening diagnostic tool to use in all patients at higher risk of lead perforation, such as those with active fixation ventricular leads, a body mass index <20, patients of older ages, and those who use temporary pacemakers and/or steroids [11]. However, the potential of TTE in visualizing the dislodgement of right and coronary sinus leads remains to be seen. In these cases, transesophageal echocardiography, or possibly intracardiac echocardiography, could be of use. 

Other techniques are available for lead perforation, but they are scarcely appealing when minimal perforation is suspected. In fact, chest radiography and fluoroscopy can be useful only when the lead migrates quite far from the heart, but in cases with minimal perforation of the heart, these tests are often non-diagnostic [1]. CT of the chest also has a major limitation: the “star artifact”, due to data loss when imaging metal implants, prevents precise location of the lead tip, and this could be an important detrimental factor in identifying minimal perforation [3]; in addition, CT implies harmful radiation exposure [13] that hampers repeatability and possible screening in asymptomatic individuals. 

Lastly, regarding the factors contributing to this complication, we think that the major factor was the use of an active fixation lead [14,15,16,17]. The rapid evolution of ICD leads has resulted in thinner active fixation leads. While these advances have made the leads more versatile, some configurations may, however, be associated with new, unforeseen complications, as demonstrated by the Riata catheter, which performed badly in terms of complications (both perforation and microperforations) [18]. Other possible mechanisms explaining active-fixation lead perforation are as follows: thin myocardial wall (our patient had a dilated cardiomyopathy that may make the right apex thinner), the overturning of the electrode’s helix and finally, the release of “hidden energy” from an overturned lead during rough manipulation during the procedure [19]. 

## 4. Conclusions

Our case shows that an optimized echocardiographic approach, with the use of non-conventional tomographic planes, can promptly detect, at the bedside, minimal cardiac catheter perforation and pericardial effusion with impending tamponade, thus driving an optimal non-invasive therapeutic strategy with fast clinical recovery.

## Figures and Tables

**Figure 1 diagnostics-10-00191-f001:**
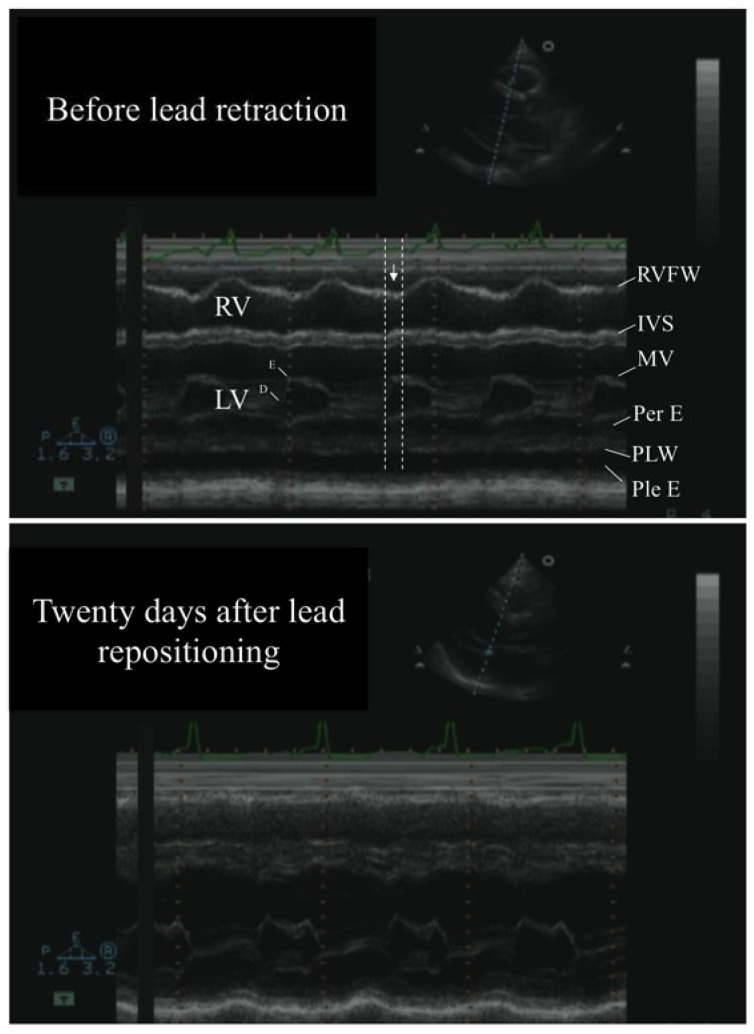
Pericardial effusion time course by transthoracic echocardiography. The upper part of the figure illustrates an m-mode at the mitral valve level before lead repositioning. The pericardial effusion (Per E) is evident, revealing a moderate-severe amount (>2 cm separation of the pericardial layers, that appears even larger toward the apex, as evident in the 2-D guiding image at the top of the screen); there is also sign of tamponade, revealed by the protodiastolic collapse of the free wall of the right ventricle (inward motion, identified by the vertical arrows), as it occurs during mitral valve opening (D–E slope) (the vertical dashed lines define the protodiastolic phase). The lower part of the figure is the same view 24 days after the lead repositioning. The pericardial effusion is totally gone. RV = right ventricle; LV = left ventricle; RVFW = right ventricle free wall; IVS = inter-ventricular septum; MV = mitral valve; Per E = pericardial effusion; PLW = posterior wall of left ventricle; Ple E = pleural effusion.

**Figure 2 diagnostics-10-00191-f002:**
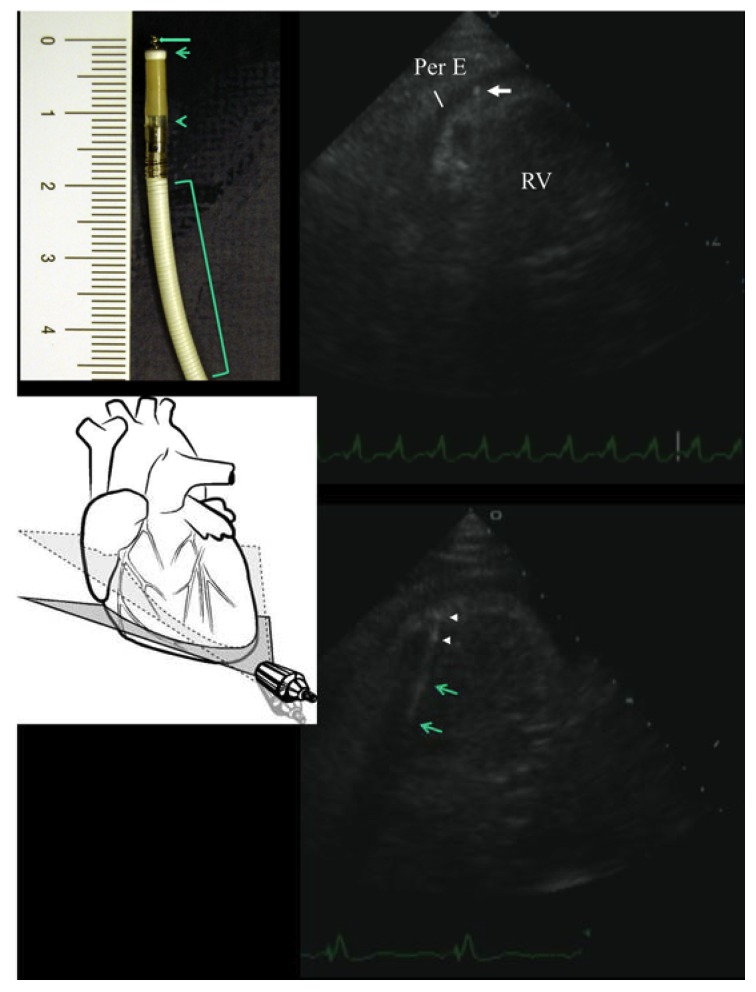
Right ventricular perforation by the ICD lead, visualized by transthoracic echocardiography, before and after lead repositioning. The upper part shows the modified 4-chamber view (the specific inclination of the tomographic plane to attain this plane is schematically shown in the diagram at the bottom left, where the dashed-line-delimited plane indicates the standard 4-chamber view plane orientation), that shows the tip of the wire (due to its minuscule structure, actually the helix) penetrating 2–3 mm into the fluid-filled pericardial space (on the upper left side, a photo of the distal part of the lead is reported, with its characteristics indicated: helix, the fixating screw (long arrow), tip electrode (short arrow), ring electrode (arrow head), and the distal defibrillator electrode (bracket)). The lower part shows the same echocardiographic view after unscrewing the helix, along with minimal lead retraction: no more protrusion of the lead into the pericardial space is visible; in addition, some hypereflective zones along the lead are visible by echo, that should correspond to the electrodes (arrow heads) and coils (arrows). RV = right ventricle; Per E = pericardial effusion.

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
