# Peer review of "Minimal Cardiac Perforation by Lead Pacemaker Complicated with Pericardial Effusion and Impending Tamponade: Optimal Management with No Pericardiocentesis Driven by Echocardiography"

_diagnostics, 2020, doi:10.3390/diagnostics10040191_

Round 1

Reviewer 1 Report

The Authors present a case of acute / subacute ICD lead perforation. 

While I strongly agree with the main message (no pericardiocentesis is needed in most of these cases) it was not avoidable due to the echocardiographic imaging plane.

The introduction shows a way to high incidence of lead perforation. Please refer to data from the Danish pacemaker registries.

The device and lead types are very old. This means, that the case is also at least 10 years old. When was this

There are several problems with the case report:

  1. The pericardial fluid was measured in ml-s. It is usually measured in mm-s in several standard planes. These numbers are not representative.
  2. The lead properties are not presented after the patient was readmitted due to the pericardial effusion and chest pain. (It was not measured for the first and second admissions). The simplest way to diagnose lead perforation is to measure lead threshold. ICD-s cannot pace in unipolar mode. For pacemaker leads it is essential to measure unipolar and bipolar thresholds.
  3. The pericardiocentesis was not postponed because the lead perforation was diagnosed, but because the patient was not in tamponade.
  4. The reason for not developing pericardial tamponade was most probably a perforation towards the pleural space. This was shown on the chest x-ray of the patient.
  5. CT cannot visualize the lead tip appropriately. True. But it is necessary to rule out other damage, like perforation to the pleural space.
  6. Discussion is unnecessary too long.

Author Response

Reviewer#1: The introduction shows a way to high incidence of lead perforation. Please refer to data from the Danisth pacemaker registries.

Response#1: we thank the reviewer for this appropriate observation; so we have cited the data of this large registry in the paper (page 1, rows 24, 25 in red).

Reviewer#2: The device and lead types are very old. This means, that the case is also at least 10 years old. When was this

Response#2: It was 8-9 years old. However we wanted to write the paper as we have noticed that the used approach was very useful also in other similar clinical settings.

Reviewer#3: The pericardial fluid was measured in ml-s. It is usually measured in mm-s in several standard planes. These numbers are not representative

Response#3: We thank again the reviewer for this constructive observation; as a rough guide, we have now used the following cut-offs for grading the size of an effusion (measured in diastole) as largely accepted (Oh J. K., Seward J. B., Tajik A. J. The Echo Manual. second edition ed. China: Lippincott Williams&Wilkins, 1999.) based on the separation of the visceral from the parietal pericardial layers:

  • Physiologic/trivial: < 5mm
  • Small : < 10 mm
  • Moderate: 10-20mm
  • Large: >20mm

In our case we reviewed the imaging studies (in particular those included in the paper as figures) and we evaluated the separation of the pericardial layers; so we replaced the ml estimation in mm of separation of the pericardial layers (marked in red in the paper, page 3, rows 77, 79, 82, 92 ). In addition we have consistently improved the Figure 1 legends adding the appropriate specifications regarding the amount of pericardial fluid (page 4, rows 102-104 in red) .

Reviewer#4: The lead properties are not presented after the patient was readmitted due to the pericardial effusion and chest pain. (It was not measured for the first and second admissions). The simplest way to diagnose lead perforation is to measure lead threshold. ICD-s cannot pace in unipolar mode. For pacemaker leads it is essential to measure unipolar and bipolar thresholds.

Response#4: We thank the reviewer for this important specification. We forgot to specify that there wasn’t any significant variation of the impedance and the other EP (electro- physiological) parameters of the lead. This is not unexpected especially in minimal perforation since normal impedance and pacing parameters did not exclude lead perforation. In fact as reported a small perforation might result in the cathode being proximal to the epicardium and the anode proximal to or within the endocardium, resulting in normal pacing parameters.

In accordance to this we have modified the text by adding the EP values at the second evaluation in the descriptive part of the case : (page 3, rows 90-92, in red); but also we have added in the discussion (page 7 , rows 180-186 in red) the explanation of the presence of normal EP parameters in association with minimal perforation and we have cited the paper to support this reasoning.

Reviewer#5: The pericardiocentesis was not postponed because the lead perforation was diagnosed, but because the patient was not in tamponade

Response#5: We thank the reviewer for this interpretation of the reported findings. However there are clear pieces of evidence that support our diagnosis of pericardial tamponade; first the clinical presentation (elevation of jugular pressure, pulsus paradox, SOB etc) second the echocardiographic evidence of large amount of pericardial fluid (>3 cm of diastolic separation of the pericardial layers) associated with protodiastolic collapse of the free wall of the right ventricle (as unequivocally evident in the Figure 1) that is a pathognomonic sign of pericardial tamponade. So the patient had an important pericardial effusion with clinical and unequivocal ultrasound signs of tamponade. For this reason the patient was immediately scheduled for urgent pericardiocentesis that was postponed after the detection of the 2 mm helix protrusion in the pericardial space. It’s evident that this decision was afforded since the tamponade was of mild entity (the patient was not in cardiogenic shock).

Reviewer#6: The reason for not developing pericardial tamponade was most probably a perforation towards the pleural space. This was shown on the chest x-ray of the patient

Response#6: Again the perforation was minimal (only the helix reached the pericardial space) and this is confirmed by the fact that by simply unscrewing the helix (a few mm retraction) we made the protruding helix in the pericardial space disappear; this drove a rapid amelioration of the symptoms (in 1 day) and totally regression of the pericardial fluid in a little more than 2 weeks. We did not show the X-ray because was unremarkable. So we do not understand this comment. The pleural effusion was indeed present (Figure 1) but was linked to venous congestion mainly related to tamponade in a patient with a dilated cardiomyopathy and atrial flutter.

Reviewer#7: CT cannot visualize the lead tip appropriately. True. But it is necessary to rule out other damage, like perforation to the pleural space

Response#7: We agree with this comment. However we were pretty sure that the perforation was minimal (thanks to echocardiography). In addition a perforation of that magnitude (reaching the pleural space) would have caused a marked increased of the impedance and the other EP parameters that was not the case. So this remarks the importance of the echocardiography that may avoid ionizing radiation procedure that can add nothing in minimal perforation.

Reviewer#8: Discussion is unnecessary too long.

Response#8: we tried to reduce the discussion but we were forced to add some other comments to respond to the reviewer comments so the total length did not change much.

Reviewer 2 Report

The presented case shows the use of a new echocardiographic plane orientation to diagnose a pericardial perforation complication of the implantation of a bi13 ventricular implantable cardioverter defibrillator. The case is of clinical importance for readers.

The figure documentation can be expanded to show more details as well as technical details of a technique, i.e. more figures and technical details on technique.

Author Response

Reviewer#1: The figure documentation can be expanded to show more details as well as technical details of a technique, i.e. more figures and technical details on technique.

Response#1: We thank the reviewer very much for this appropriate observation; we agree with him that a new approach has to be fully supported with illustration and schematics. We tried in fact, to explain the off-axis ultrasound approach with a cartoon (Figure 2 top left) that should explain the tomographic plane orientation for visualizing the tip of the lead in the pericardial space. Unfortunately we can’t add other material. We reviewed the patient’s clinical file but we didn’t find anything else useful for this purpose.

Round 2

Reviewer 1 Report

One last issue:

I would change the title. Cardiac tamponade means a situation where you must perform pericardiocentesis or surgery. That is why we call it tampoande (low blood pressure, cardiogenic shock). If there is no need for pericardiocentesis (blood pressure and urine output OK) it is called pericardial effusion. 

Please also change this in the text to have a coherent nomenclature. 

Author Response

Reviewer comment: I would change the title. Cardiac tamponade means a situation where you must perform pericardiocentesis or surgery. That is why we call it tampoande (low blood pressure, cardiogenic shock). If there is no need for pericardiocentesis (blood pressure and urine output OK) it is called pericardial effusion. Please also change this in the text to have a coherent nomenclature. 

Answer: I’d like to thank the reviewer for his/her genuine effort to improve the manuscript. In accordance with the literature [1] cardiac tamponade can be defined clinically or echocardiographically. Clinically, cardiac tamponade is defined as the presence of pericardial fluid causing circulatory collapse, as indicated by blood pressure and perfusion status. Echocardiographically, cardiac tamponade shows diastolic collapse of the right ventricle, and is usually referred to as impending tamponade. So our case certainly belongs to the second category of tamponade (impending tamponade: signs of echocardiographic signs of tamponade without overt cardiac hypotension); impending because if the situation remains the same, the clinical scenario gets worse in short and eventually hypotension with hypoperfusion (central shock) takes over.

In accordance, as suggested, we have modified:

Title, row 3 in orange;

Introduction , rows 39-40

Case, row 124

Conclusion, row 204

References, we have added ref 6, in orange

  1. Price AS, Leech SJ, Sierzenski PR. Impending cardiac tamponade: a case report highlighting the value of bedside echocardiography. The Journal of emergency medicine. 2006;30(4):415-9. 10.1016/j.jemermed.2005.07.012